# Welfare Guarantees from Data

**Darrell Hoy**
University of Maryland
darrell.hoy@gmail.com

**Denis Nekipelov**
University of Virginia
denis@virginia.edu

**Vasilis Syrgkanis**
Microsoft Research
vasy@microsoft.com

## Abstract

Analysis of efficiency of outcomes in game theoretic settings has been a main item of study at the intersection of economics and computer science. The notion of the price of anarchy takes a worst-case stance to efficiency analysis, considering instance independent guarantees of efficiency. We propose a data-dependent analog of the price of anarchy that refines this worst-case assuming access to samples of strategic behavior. We focus on auction settings, where the latter is non-trivial due to the private information held by participants. Our approach to bounding the efficiency from data is robust to statistical errors and mis-specification. Unlike traditional econometrics, which seek to learn the private information of players from observed behavior and then analyze properties of the outcome, we directly quantify the inefficiency without going through the private information. We apply our approach to datasets from a sponsored search auction system and find empirical results that are a significant improvement over bounds from worst-case analysis.

## 1 Introduction

A major field at the intersection of economics and computer science is the analysis of the efficiency of systems under strategic behavior. The seminal work of [6, 11] triggered a line of work on quantifying the inefficiency of computer systems, ranging from network routing, resource allocation and more recently auction marketplaces [10]. However, the notion of the price of anarchy suffers from the pessimism of worst-case analysis. Many systems can be inefficient in the worst-case over parameters of the model, but might perform very well for the parameters that arise in practice.

Due to the large availability of datasets in modern economic systems, we propose a data-dependent analog of the price of anarchy, which assumes access to a sample of strategic behavior from the system. We focus our analysis on auction systems where the latter approach is more interesting due to the private information held by the participants of the system, i.e. their private value for the item at sale. Since efficiency is a function of these private parameters, quantifying the inefficiency of the system from samples of strategic behavior is non-trivial. The problem of estimation of the inefficiency becomes an econometric problem where we want to estimate a function of hidden variables from observed strategic behavior. The latter is feasible under the assumption that the observed behavior is the outcome of an equilibrium of the strategic setting, which connects observed behavior to unobserved private information.

Traditional econometric approaches to auctions [3, 8], address such questions by attempting to exactly pin-point the private parameters from the observed behavior and subsequently measuring the quantities of interest, such as the efficiency of the allocation. The latter approach is problematic in complex auction systems for two main reasons: (i) it leads to statistical inefficiency, (ii) it requires strong conditions on the connection between observed behavior and private information. Even for a single-item first-price auction, uniform estimation of the private value of a player from $T$ samples of observed bids, can only be achieved at $O(T^{1/3})$-rates [3]. Moreover, uniquely identifying the private information from the observed behavior, requires a one-to-one mapping between the two

quantities. The latter requires strong assumptions on the distribution of private parameters and can only be applied to simple auction rules.

Our approach bridges the gap between worst-case price of anarchy analysis and statistically and modeling-wise brittle econometric analysis. We provide a data-dependent analog of recent techniques for quantifying the worst-case inefficiency in auctions [13, 4, 10], that do not require characterization of the equilibrium structure and which directly quantify the inefficiency through best-response arguments, without the need to pin-point the private information. Our approach makes minimal assumptions on the distribution of private parameters and on the auction rule and achieves $\tilde{O}(\sqrt{T})$-rates of convergence for many auctions used in practice, such as the Generalized Second Price (GSP) auction [2, 14]. We applied our approach to a real world dataset from a sponsored search auction system and we portray the optimism of the data-dependent guarantees as compared to their worst-case counterparts [1].

## 2   Preliminaries

We consider the single-dimensional mechanism design setting with $n$ bidders. The mechanism designer wants to allocate a unit of good to the bidders, subject to some feasibility constraint on the vector of allocations $(x_1, \ldots, x_n)$. Let $\mathcal{X}$ be the space of feasible allocations. Each bidder $i$ has a private value $v_i \in [0, H]$ per-unit of the good, and her utility when she gets allocation $x_i$ and is asked to make a payment $p_i$ is $v_i \cdot x_i - p_i$. The value of each bidder is drawn independently from distribution with CDF $F_i$, supported in $V_i \subseteq \mathbb{R}_+$ and let $\mathbf{F} = \times_i F_i$ be the joint distribution.

An auction $A$ solicits a bid $b_i \in \mathcal{B}$ from each bidder $i$ and decides on the allocation vector based on an allocation rule $\mathbf{X} : \mathcal{B}^n \to \mathcal{X}$ and a payment rule $\mathbf{p} : \mathcal{B}^n \to \mathbb{R}^n$. For a vector of values and bids, the utility of a bidder is:

$$U_i(\mathbf{b}; v_i) = v_i \cdot X_i(\mathbf{b}) - P_i(\mathbf{b}). \tag{1}$$

A strategy $\sigma_i : V_i \to \mathcal{B}$, for each bidder $i$, maps the value of the bidder to a bid. Given an auction $A$ and distribution of values $\mathbf{F}$, a strategy profile $\sigma$ is a *Bayes-Nash Equilibrium* (BNE) if each bidder $i$ with any value $v_i \in V_i$ maximizes her utility in expectation over her opponents bids, by bidding $\sigma_i(v_i)$.

The welfare of an auction outcome is the expected utility generated for all the bidders, plus the revenue of the auctioneer, which due to the form of bidder utilities boils down to being the total value that the bidders get from the allocation. Thus the expected utility of a strategy profile $\sigma$ is

$$\text{WELFARE}(\sigma; \mathbf{F}) = \mathbf{E}_{\mathbf{v} \sim \mathbf{F}} \left[ \sum_{i \in [n]} v_i \cdot X_i(\sigma(\mathbf{v})) \right] \tag{2}$$

We denote with $\text{OPT}(\mathbf{F})$ the expected optimal welfare: $\text{OPT}(\mathbf{F}) = \mathbf{E}_{\mathbf{v} \sim \mathbf{F}}[\max_{\mathbf{x} \in \mathcal{X}} \sum_{i \in [n]} v_i \cdot x_i]$.

**Worst-case Bayes-Nash price of anarchy.**   The Bayesian price of anarchy of an auction is defined as the worst-case ratio of welfare in the optimal auction to the welfare in a Bayes-Nash equilibrium of the original auction, taken over all value distributions and over all equilibria. Let $BNE(A, \mathbf{F})$ be the set of Bayes-Nash equilibria of an auction $A$, when values are drawn from distributions $\mathbf{F}$. Then:

$$\text{POA} = \sup_{\mathbf{F}, \sigma \in BNE(\mathbf{F})} \frac{\text{OPT}(\mathbf{F})}{\text{WELFARE}(\sigma; \mathbf{F})} \tag{3}$$

## 3   Distributional Price of Anarchy: Refining the POA with Data

We will assume that we observe $T$ samples $\mathbf{b}^{1:T} = \{\mathbf{b}^1, \ldots, \mathbf{b}^T\}$ of bid profiles from running $T$ times an auction $A$. Each bid profile $\mathbf{b}^t$ is drawn i.i.d. based on an unknown Bayes-Nash equilibrium $\sigma$ of the auction, i.e.: let $\mathcal{D}$ denote the distribution of the random variable $\sigma(\mathbf{v})$, when $\mathbf{v}$ is drawn from $\mathbf{F}$. Then $b^t$ are i.i.d. samples from $\mathcal{D}$. Our goal is to refine our prediction on the efficiency of the auction and compute a bound on the price of anarchy of the auction conditional on the observed data set. More formally, we want to derive statements of the form: conditional on $\mathbf{b}^{1:T}$, with probability at least $1 - \delta$: $\text{WELFARE}(\sigma; \mathbf{F}) \geq \frac{1}{\hat{\rho}}\text{OPT}(\mathbf{F})$, where $\hat{\rho}$ is the empirical analogue of the worst-case price of anarchy ratio.

**Infinite data limit**    We will tackle this question in two steps, as is standard in estimation theory. First we will look at the infinite data limit where we know the actual distribution of equilibrium bids $\mathcal{D}$. We define a notion of price of anarchy that is tailored to an equilibrium bid distribution, which we refer to as the *distributional price of anarchy*. In Section 4 we give a distribution-dependent upper bound on this ratio for any single-dimensional auction. Subsequently, in Section 5, we show how one can estimate this upper bound on the distributional price of anarchy from samples.

Given a value distribution $\mathbf{F}$ and an equilibrium $\sigma$, let $D(\mathbf{F}, \sigma)$ denote the resulting equilibrium bid distribution. We then define the *distributional price of anarchy* as follows:

**Definition 1** (Distributional Price of Anarchy)**.** *The* distributional price of anarchy DPoA$(\mathcal{D})$ *of an auction $A$ and a distribution of bid profiles $\mathcal{D}$, is the worst-case ratio of welfare in the optimal allocation to the welfare in an equilibrium, taken over all distributions of values and all equilibria that could generate the bid distribution $\mathcal{D}$:*

$$\text{DPoA}(\mathcal{D}) = \sup_{\mathbf{F}, \sigma \in BNE(\mathbf{F}) \text{ s.t. } D(\mathbf{F},\sigma)=\mathcal{D}} \frac{\text{OPT}(\mathbf{F})}{\text{WELFARE}(\sigma; \mathbf{F})} \qquad (4)$$

This notion has nothing to do with sampled data-sets, but rather is a hypothetical worst-case quantity that we could calculate had we known the true bid generating distribution $\mathcal{D}$.

**What does the extra information of knowing $\mathcal{D}$ give us?**    To answer this question, we first focus on the optimization problem each bidder faces. At any Bayes-Nash equilibrium each player must be best-responding in expectation over his opponent bids. Observe that if we know the rules of the auction and the equilibrium distribution of bids $\mathcal{D}$, then the expected allocation and payment function of a player as a function of his bid are uniquely determined:

$$x_i(b; \mathcal{D}) = \mathbf{E}_{\mathbf{b}_{-i} \sim \mathcal{D}_{-i}}[X_i(b, \mathbf{b}_{-i})] \qquad p_i(b; \mathcal{D}) = \mathbf{E}_{\mathbf{b}_{-i} \sim \mathcal{D}_{-i}}[P_i(b, \mathbf{b}_{-i})]. \qquad (5)$$

Importantly, these functions do not depend on the distribution of values $\mathbf{F}$, other than through the distribution of bids $\mathcal{D}$. Moreover, the expected revenue of the auction is also uniquely determined:

$$\text{REV}(\mathcal{D}) = \mathbf{E}_{\mathbf{b} \sim \mathcal{D}}\left[\sum_i \mathbf{P}_i(\mathbf{b})\right], \qquad (6)$$

Thus when bounding the *distributional price of anarchy*, we can assume that these functions and the expected revenue are known. The latter is unlike the standard price of anarchy analysis, which essentially needs to take a worst-case approach to these quantities.

**Shorthand notation**    Through the rest of the paper we will fix the distribution $\mathcal{D}$. Hence, for brevity we omit it from notation, using $x_i(b)$, $p_i(b)$ and REV instead of $x_i(b; \mathcal{D})$, $p_i(b; \mathcal{D})$ and REV$(\mathcal{D})$.

## 4   Bounding the Distributional Price of Anarchy

We first upper bound the *distributional price of anarchy* via a quantity that is relatively easy to calculate as a function of the bid distribution $\mathcal{D}$ and hence will also be rather straightforward to estimate from samples of $\mathcal{D}$, which we defer to the next section. To give intuition about the upper bound, we start with a simple but relevant example of bounding the distributional price of anarchy in the case when the auction $A$ is the single-item first price auction. We then generalize the approach to any auction $A$.

### 4.1   Example: Single-Item First Price Auction

In a single item first price auction, the designer wants to auction a single indivisible good. Thus the space of feasible allocations $\mathcal{X}$, are ones where only one player gets allocation $x_i = 1$ and other players get allocation $0$. The auctioneer solicits bids $b_i$ from each bidder and allocates the good to the highest bidder (breaking ties lexicographically), charging him his bid. Let $\mathcal{D}$ be the equilibrium distribution of bids and let $G_i$ be the CDF of the bid of player $i$. For simplicity we assume that $G_i$ is continuous (i.e. the distribution is atomless). Then the expected allocation of a player $i$ from submitting a bid $b$ is equal to $x_i(b) = G_{-i}(b) = \prod_{j \neq i} G_j(b)$ and his expected payment is $p_i(b) = b \cdot x_i(b)$, leading to expected utility: $u_i(b; v_i) = (v_i - b)G_{-i}(b)$.

The quantity DPOA is a complex object as it involves the structure of the set of equilibria of the given auction. The set of equilibria of a first price auction when bidders values are drawn from different distributions is an horrific object.[1] However, we can upper bound this quantity by a much simpler data-dependent quantity by simply invoking the fact that under any equilibrium bid distribution no player wants to deviate from his equilibrium bid. Moreover, this data-dependent quantity can be much better than its worst-case counterpart used in the existing literature on the price of anarchy.

**Lemma 1.** *Let $A$ be the single item first price auction and let $\mathcal{D}$ be the equilibrium distribution of bids, then* $\mathrm{DPOA}(\mathcal{D}) \leq \frac{\mu(\mathcal{D})}{1-e^{-\mu(\mathcal{D})}}$, *where* $\mu(\mathcal{D}) = \frac{\max_{i\in[n]} \mathbb{E}_{\mathbf{b}_{-i}\sim\mathcal{D}_{-i}}[\max_{j\neq i} b_j]}{\mathbb{E}_{\mathbf{b}\sim\mathcal{D}}[\max_{i\in[n]} b_i]}$.

*Proof.* Let $G_i$ be the CDF of the bid of each player under distribution $\mathcal{D}$. Moreover, let $\sigma$ denote the equilibrium strategy that leads to distribution $\mathcal{D}$. By the equilibrium condition, we know that for all $v_i \in V_i$ and for all $b' \in \mathcal{B}$,

$$u_i(\sigma_i(v_i); v_i) \geq u_i(b'; v_i) = (v_i - b') \cdot G_{-i}(b'). \tag{7}$$

We will give a special deviating strategy used in the literature [13], that will show that either the players equilibrium utility is large or the expected maximum other bid is high. Let $T_i$ denote the expected maximum other bid which can be expressed as $T_i = \int_0^\infty 1 - G_{-i}(z)dz$. We consider the randomized deviation where the player submits a randomized bid in $z \in [0, v_i(1-e^{-\mu})]$ with PDF $f(z) = \frac{1}{\mu(v_i-z)}$. Then the expected utility from this deviation is:

$$\mathbb{E}_{b'}[u_i(b'; v_i)] = \int_0^{v_i(1-e^{-\mu})} (v_i - z) \cdot G_{-i}(z)f(z)dz = \frac{1}{\mu} \int_0^{v_i(1-e^{-\mu})} G_{-i}(z)dz \tag{8}$$

Adding the quantity $\frac{1}{\mu}\int_0^{v_i(1-e^{-\mu})}(1 - G_{-i}(z))dz \leq \frac{1}{\mu}T_i$ on both sides, we get: $\mathbb{E}_{b'}[u_i(b'; v_i)] + \frac{1}{\mu}T_i \geq v_i\frac{1-e^{-\mu}}{\mu}$. Invoking the equilibrium condition we get: $u_i(\sigma_i(v_i); v_i) + \frac{1}{\mu}T_i \geq v_i\frac{1-e^{-\mu}}{\mu}$. Subsequently, for any $x_i^* \in [0,1]$:

$$u_i(\sigma_i(v_i); v_i) + \frac{1}{\mu}T_i \cdot x_i^* \geq v_i \cdot x_i^* \frac{1-e^{-\mu}}{\mu}. \tag{9}$$

If $x_i^*$ is the expected allocation of player $i$ under the efficient allocation rule $X_i^*(\mathbf{v}) \equiv 1\{v_i = \max_j v_j\}$, then taking expectation of Equation (9) over $v_i$ and adding across all players we get:

$$\sum_i \mathbb{E}_{v_i}[u_i(\sigma_i(v_i); v_i)] + \frac{1}{\mu}\mathbb{E}_{\mathbf{v}}\left[\sum_i T_i X_i^*(\mathbf{v})\right] \geq \mathrm{OPT}(\mathbf{F})\frac{1-e^{-\mu}}{\mu} \tag{10}$$

The theorem then follows by invoking the fact that for any feasible allocation $x$: $\sum_i T_i \cdot x_i \leq \max_i T_i = \mu(\mathcal{D})\mathrm{REV}(\mathcal{D})$, using the fact that expected total agent utility plus total revenue at equilibrium is equal to expected welfare at equilibrium and setting $\mu = \mu(\mathcal{D})$. ∎

**Comparison with worst-case POA** In the worst-case, $\mu(\mathcal{D})$ is upper bounded by 1, leading to the well-known worst-case price of anarchy ratio of the single-item first price auction of $(1 - 1/e)^{-1}$, irrespective of the bid distribution $\mathcal{D}$. However, if we know the distribution $\mathcal{D}$ then we can explicitly estimate $\mu$, which can lead to a much better ratio (see Figure 1). Moreover, observe that even if we had samples from the bid distribution $\mathcal{D}$, then estimating $\mu(\mathcal{D})$ is very easy as it corresponds to the ratio of two expectations, each of which can be estimating to within an $O(\frac{1}{\sqrt{T}})$ error by a simple average and using standard concentration inequalities. Even thought this improvement, when compared to the worst-case bound might not be that drastic in the first price auction, the extension of the analysis in the next section will be applicable even to auctions where the analogue of the quantity $\mu(\mathcal{D})$ is not even bounded in the worst-case. In those settings, the empirical version of the price of anarchy analysis is of crucial importance to get any efficiency bound.

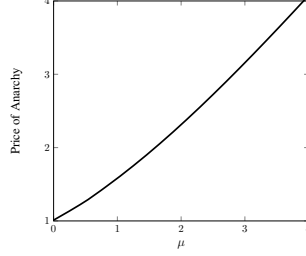

**Figure 1:** The upper bound on the distributional price of anarchy of an auction $\frac{\mu(\mathcal{D})}{1-e^{-\mu(\mathcal{D})}}$ as a function of $\mu(\mathcal{D})$.

**Comparison with value inversion approach**    Apart from being just a primer to our main general result in the next section, the latter result about the data-dependent efficiency bound for the first price auction, is itself a contribution to the literature. It is notable to compare the latter result with the standard econometric approach to estimating values in a first price auction pioneered by [3] (see also [8]). Traditional non-parametric auction econometrics use the equilibrium best response condition to pin-point the value of a player from his observed bid, by what is known as value inversion. In particular, if the function: $u_i(b'; v_i) = (v_i - b') \cdot G_{-i}(b')$ has a unique maximum for each $v_i$ and this maximum is strictly monotone in $v_i$, then given the equilibrium bid of a player $b_i$ and given a data distribution $\mathcal{D}$ we can reverse engineer the value $v_i(b_i)$ that the player must have. Thus if we know the bid distribution $\mathcal{D}$ we can calculate the equilibrium welfare as $\mathbb{E}_{\mathbf{b} \sim \mathcal{D}}\left[\sum_i v_i(b_i) \cdot X_i(\mathbf{b})\right]$. Moreover, we can calculate the expected optimal welfare as: $\mathbb{E}_{\mathbf{b} \sim \mathcal{D}}\left[\max_i v_i(b_i)\right]$. Thus we can pin-point the distributional price of anarchy.

However, the latter approach suffers from two main drawbacks: (i) estimating the value inversion function $v_i(\cdot)$ uniformly over $b$ from samples, can only happen at very slow rates that are at least $O(1/T^{1/3})$ and which require differentiability assumptions from the value and bid distribution as well as strong conditions that the density of the value distribution is bounded away from zero in all the support (with this lower bound constant entering the rates of convergence), (ii) the main assumption of the latter approach is that the optimal bid is an invertible function and that given a bid there is a single value that corresponds to that bid. This assumption might be slightly benign in a single item first price auction, but becomes a harsher assumption when one goes to more complex auction schemes. Our result in Lemma 1 suffers neither of these drawbacks: it admits fast estimation rates from samples, makes no assumption on properties of the value and bid distribution and does not require invertibility of the best-response correspondence. Hence it provides an upper bound on the distributional price of anarchy that is statistically robust to both sampling and mis-specification errors. The robustness of our approach comes with the trade-off that we are now only estimating a bound on the efficiency of the outcome, rather than exactly pinpointing it.

## 4.2   Generalizing to any Single-Dimensional Auction Setting

Our analysis on DPoA is based on the reformulation of the auction rules as an equivalent *pay-your-bid* auction and then bounding the price of anarchy as a function of the ratio of how much a player needs to pay in an equivalent *pay-your-bid* auction, so as to acquire his optimal allocation vs. how much revenue is the auctioneer collecting. For any auction, we can re-write the expected utility of a bid $b$:

$$u_i(b; v_i) = x_i(b)\left(v_i - \frac{p_i(b)}{x_i(b)}\right) \tag{11}$$

This can be viewed as the same form of utility if the auction was a *pay-your-bid* auction and the player submitted a bid of $\frac{p_i(b)}{x_i(b)}$. We refer to this term as the price-per-unit and denote it $\mathrm{ppu}(b) = \frac{p_i(b)}{x_i(b)}$. Our analysis will be based on the *price-per-unit* allocation rule $\tilde{x}(\cdot)$, which determines the expected allocation of a player as a function of his price-per-unit. Given this notation, we can re-write the utility that an agent achieves if he submits a bid that corresponds to a price-per-unit of $z$ as: $\tilde{u}_i(z; v_i) = \tilde{x}(z)(v_i - z)$. The latter is exactly the form of a *pay-your-bid* auction.

Our upper bound on the DPOA, will be based on the inverse of the PPU allocation rule; let $\tau_i(z) = \tilde{x}_i^{-1}(z)$ be the price-per-unit of the cheapest bid that achieves allocation at least $z$. More formally,

$\tau_i(z) = \min_{b|x_i(b) \geq z}\{\text{ppu}(b)\}$. For simplicity, we assume that any allocation $z \in [0,1]$ is achieveable by some high enough bid $b$.[2] Given this we can define the threshold for an allocation:

**Definition 2** (Average Threshold). *The* average threshold *for agent $i$ is*

$$T_i = \int_0^1 \tau_i(z)\, dz \tag{12}$$

In Figures 3 and 2 we provide a pictorial representation of these quantities. Connecting with the previous section, for a first price auction, the price-per-unit function is $\text{ppu}(b) = b$, the price-per-unit allocation function is $\tilde{x}_i(b) = G_{-i}(b)$ and the threshold function is $\tau_i(z) = G_{-i}^{-1}(z)$. The average threshold $T_i$ is equal to $\int_0^1 G_{-i}^{-1}(z)dz = \int_0^\infty 1 - G_{-i}(b)db$, i.e. the expected maximum other bid.

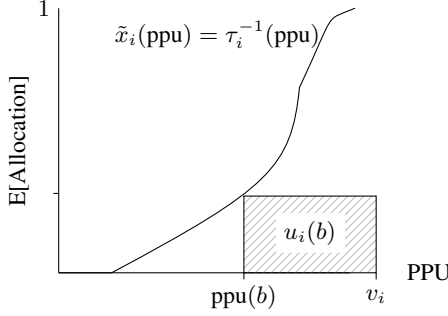

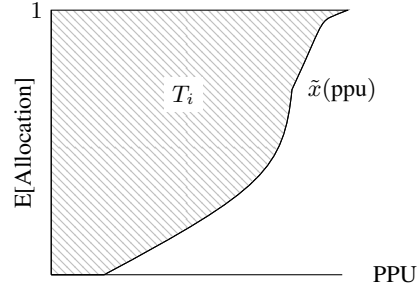

**Figure 2:** For any bid $b$ with PPC $\text{ppu}(b)$, the area of a rectangle between $(\text{ppu}(b), \tilde{x}_i(\text{ppu}(b)))$ and $(v_i, 0)$ on the bid allocation rule is the expected utility $u_i(b)$. The BNE action $b^*$ is chosen to maximize this area.

**Figure 3:** The average threshold is the area to the left of the price-per-unit allocation rule, integrate from 0 to 1.

We now give our main theorem, which is a distribution-dependent bound on DPoA, that is easy to compute give $\mathcal{D}$ and which can be easily estimated from samples of $\mathcal{D}$. This theorem is a generalization of Lemma 1 in the previous section.

**Theorem 2** (Distributional Price of Anarchy Bound). *For any auction $A$ in a single dimensional setting and for any bid distribution $\mathcal{D}$, the distributional price of anarchy is bounded by $\text{DPoA}(\mathcal{D}) \leq \frac{\mu(\mathcal{D})}{1 - e^{-\mu(\mathcal{D})}}$, where $\mu(\mathcal{D}) = \frac{\max_{x \in \mathcal{X}} \sum_{i=1}^n T_i \cdot x_i}{\text{REV}(\mathcal{D})}$.*

Theorem 2 provides our main method for bounding the distributional price of anarchy. All we need is to compute the revenue REV of the auction and the quantity:

$$\mathbf{T} = \max_{x \in \mathcal{X}} \sum_{i=1}^n T_i \cdot x_i, \tag{13}$$

under the given bid distribution $\mathcal{D}$. Both of these are uniquely defined quantities if we are given $\mathcal{D}$. Moreover, once we compute $T_i$, the optimization problem in Equation (13) is simply a welfare maximization problem, where each player's value per-unit of the good is $T_i$. Thus, the latter can be solved in polynomial time, whenever the welfare maximization problem over the feasible set $\mathcal{X}$ is polynomial-time solvable.

Theorem 2 can be viewed as a bid distribution-dependent analogue of the *revenue covering framework* [4] and of the smooth mechanism framework [13]. In particular, the quantity $\mu(\mathcal{D})$ is the data-depenent analogue of the worst-case $\mu$ quantity used in the definition of $\mu$-revenue covering in [4] and is roughly related to the $\mu$ quantity used in the definition of a $(\lambda, \mu)$-smooth mechanism in [13].

## 5   Distributional Price of Anarchy Bound from Samples

In the last section, we assumed we were given distribution $\mathcal{D}$ and hence we could compute the quantity $\mu = \frac{\mathbf{T}}{\text{REV}}$, which gave an upper bound on the DPoA. We now show how we can estimate this

quantity $\mu$ when given access to i.i.d. samples $\mathbf{b}^{1:T}$ from the bid distribution $\mathcal{D}$. We will separately estimate $\mathbf{T}$ and REV. The latter is simple expectation and thereby can be easily estimated by an average at $\frac{1}{\sqrt{T}}$ rates. For the former we first need to estimate $T_i$ for each player $i$, which requires estimation of the allocation and payment functions $x_i(\cdot; \mathcal{D})$ and $p_i(\cdot; \mathcal{D})$.

Since both of these functions are expected values over the equilibrium bids of opponents, we will approximate them by their empirical analogues:

$$\widehat{x_i}(b) = \frac{1}{T} \sum_{t=1}^{T} X_i(b, \mathbf{b}_{-i}^t) \qquad\qquad \widehat{p_i}(b) = \frac{1}{T} \sum_{t=1}^{T} P_i(b, \mathbf{b}_{-i}^t). \tag{14}$$

To bound the estimation error of the quantities $\hat{T}_i$ produced by using the latter empirical estimates of the allocation and payment function, we need to provide a uniform convergence property for the error of these functions over the bid $b$.

Since $b$ takes values in a continuous interval, we cannot simply apply a union bound. We need to make assumptions on the structure of the class of functions $\mathcal{F}_{X_i} = \{X_i(b, \cdot) : b \in \mathcal{B}\}$ and $\mathcal{F}_{P_i} = \{P_i(b, \cdot) : b \in \mathcal{B}\}$, so as uniformly bound their estimation error. For this we resort to the technology of Rademacher complexity. For a generic class of functions $\mathcal{F}$ and a sequence of random variables $Z^{1:T}$, the Rademacher complexity is defined as:

$$\mathcal{R}_T(\mathcal{F}, Z^{1:T}) = \mathop{\mathbb{E}}_{\sigma^{1:T}} \left[ \sup_{f \in \mathcal{F}} \frac{1}{T} \sum_{t=1}^{T} \sigma^t f(Z^t) \right]. \tag{15}$$

where each $\sigma^t \in \{\pm 1/2\}$ is an i.i.d. Rademacher random variable, which takes each of those values with equal probabilities. The following well known theorem will be useful in our derivations:

**Theorem 3** ([12]). *Suppose that for any sample $Z^{1:T}$ of size $T$, $\mathcal{R}_T(\mathcal{F}, Z^{1:T}) \leq \mathcal{R}_T$ and suppose that functions in $\mathcal{F}$ take values in $[0, H]$. Then with probability $1 - \delta$:*

$$\sup_{f \in \mathcal{F}} \left| \frac{1}{T} \sum_{t=1}^{T} f(Z_t) - \mathbb{E}[f(Z)] \right| \leq 2\mathcal{R}_T + H \sqrt{\frac{2 \log(4/\delta)}{T}} \tag{16}$$

This Theorem reduces our uniform error problem to bounding the Rademacher complexity of classes $\mathcal{F}_{X_i}$ and $\mathcal{F}_{P_i}$, since we immediately have the following corollary (where we also use that the allocation functions lie in $[0, 1]$ and the payment functions lie in $[0, H]$):

**Corollary 4.** *Suppose that for any sample $\mathbf{b}^{1:T}$ of size $T$, the Rademacher complexity of classes $\mathcal{F}_{X_i}$ and $\mathcal{F}_{P_i}$ is at most $\mathcal{R}_T$. Then with probability $1 - \delta/2$, both $\sup_{b \in \mathcal{B}} |\widehat{x_i}(b) - x_i(b)|$ and $\sup_{b \in \mathcal{B}} |\widehat{p_i}(b) - p_i(b)|$ are at most $2\mathcal{R}_T + H \sqrt{2 \log(4/\delta)/T}$.*

We now provide conditions under which the Rademacher complexity of these classes is $\tilde{O}(1/\sqrt{T})$.

**Lemma 5.** *Suppose that $\mathcal{B} = [0, B]$ and for each bidder $i$ and each $b_i \in \mathcal{B}$, the functions $X_i(b_i, \cdot) : [0, B]^{n-1} \mapsto [0, 1]$ and $P_i(b, \cdot) : [0, B]^{n-1} \mapsto [0, H]$ can be computed as finite superposition of (i) coordinate-wise multiplication of bid vectors $\mathbf{b}_{-i}$ with constants; (ii) comparison indicators $\mathbf{1}\{\cdot > \cdot\}$ of coordinates or constants; (iii) pairwise addition $\cdot + \cdot$ of coordinates or constants. The Rademacher complexity for both classes on a sample of size $T$ is $O\left( \sqrt{\log(T)/T} \right)$.*

The proof of this Lemma follows by standard arguments of Rademacher calculus, together with VC arguments on the class of pairwise comparisons. Those arguments can be found in [5, Lemma 9.9] and [9, Lemma 11.6.28]. Thereby, we omit its proof. The assumptions of Lemma 5 can be directly verified, for instance, for the sponsored search auctions where the constants that multiply each bid correspond to quality factors of the bidders, e.g. as in [2] and [14] and then the allocation and the payment is a function of the rank of the weighted bid of a player. In that case the price and the allocation rule are determined solely by the ranks and the values of the score-weighted bids $\gamma_i b_i$, as well as the position specific quality factors $\alpha_j$, for each position $j$ in the auction.

Next we turn to the analysis of the estimation errors on quantities $T_i$. We consider the following plug-in estimator for $T_i$: We consider the empirical analog of function $\tau_i(\cdot)$ by $\widehat{\tau}_i(z) = \inf_{b \in [0, B], \widehat{x_i}(b) \geq z} \frac{\widehat{p_i}(b)}{\widehat{x_i}(b)}$.

Then the empirical analog of $T_i$ is obtained by:

$$\widehat{T}_i = \int_0^1 \widehat{\tau}_i(z) \, dz. \tag{17}$$

To bound the estimation error of $\widehat{T}_i$, we need to impose an additional condition that ensures that any non-zero allocation requires the payment from the bidder at least proportional to that allocation.

**Assumption 6.** *We assume that $p_i(x_i^{-1}(\cdot))$ is Lipschitz-continuous and that the mechanism is worst-case interim individually rational, i.e. $p_i(b) \leq H \cdot x_i(b)$.*

Under this assumption we can establish that $\tilde{O}(\sqrt{T})$ rates of convergence of $\widehat{T}_i$ to $T_i$ and of the empirical analog $\widehat{\mathbf{T}} = \max_{x \in \mathcal{X}} \sum_{i=1}^n \widehat{T}_i \cdot x_i$ of the optimized threshold to $\mathbf{T}$ as well as the empirical analog $\widehat{\text{REV}}$ of the revenue to REV. Thus the quantity $\hat{\mu} = \frac{\widehat{\mathbf{T}}}{\widehat{\text{REV}}}$, will also converge to $\mu = \frac{\mathbf{T}}{\text{REV}}$ at that rate. This implies the following final conclusion of this section.

**Theorem 7.** *Under Assumption 6 and the premises of Lemma 5, with probability $1 - \delta$:*

$$\frac{\text{OPT}(\mathbf{F})}{\text{WELFARE}(\sigma; \mathbf{F})} \leq \frac{\widehat{\mu}}{1 - e^{-\widehat{\mu}}} + \tilde{O}\left(n \max\{L, H\} \sqrt{\frac{H \log(n/\delta)}{T}}\right) \tag{18}$$

## 6 Sponsored Search Auction: Model, Methodology and Data Analysis

We consider a position auction setting where $k$ ordered positions are assigned to $n$ bidders. An outcome $m$ in a position auction is an allocation of positions to bidders. $m(j)$ denotes the bidder who is allocated position $j$; $m^{-1}(i)$ refers to the position assigned to bidder $i$. When bidder $i$ is assigned to slot $j$, the probability of click $c_{i,j}$ is the product of the click-through-rate of the slot $\alpha_j$ and the quality score of the bidder, $\gamma_i$, so $c_{i,j} = \alpha_j \gamma_i$ (in the data the quality scores for each bidder are varying across different auctions and we used the average score as a proxy for the score of a bidder). Each advertiser has a value-per-click (VPC) $v_i$, which is not observed in the data and which we assume is drawn from some distribution $F_i$. Our benchmark for welfare will be the welfare of the auction that chooses a feasible allocation to maximize the welfare generated, thus $\text{OPT} = \mathbf{E}_\mathbf{v}[\max_m \sum_i \gamma_i \alpha_{m^{-1}(i)} v_i]$.

We consider data generated by advertisers repeatedly participating in a sponsored search auction. The mechanism that is being repeated at each stage is an instance of a generalized second price auction triggered by a search query. The rules of each auction are as follows: Each advertiser $i$ is associated with a click probability $\gamma_i$ and a scoring coefficient $s_i$ and is asked to submit a bid-per-click $b_i$. Advertisers are ranked by their rank-score $q_i = s_i \cdot b_i$ and allocated positions in decreasing order of rank-score as long as they pass a rank-score reserve $r$. All the mentioned sets of parameters $\theta = (\mathbf{s}, \alpha, \gamma, r)$ and the bids $\mathbf{b}$ are observable in the data.

We will denote with $\pi_{\mathbf{b},\theta}(j)$ the bidder allocated in slot $j$ under a bid profile $\mathbf{b}$ and parameter profile $\theta$. We denote with $\pi_{\mathbf{b},\theta}^{-1}(i)$ the slot allocated to bidder $i$. If advertiser $i$ is allocated position $j$, then he pays only when he is clicked and his payment, i.e. his cost-per-click is the minimal bid he had to place to keep his position, which is: $\text{cpc}_{ij}(\mathbf{b}; \theta) = \frac{\max\left\{s_{\pi_{\mathbf{b},\theta}(j+1)} \cdot b_{\pi_{\mathbf{b},\theta}(j+1)}, r\right\}}{s_i}$. Mapping this setting to our general model, the allocation function of the auction is $X_i(\mathbf{b}) = \alpha_{\pi_{\mathbf{b},\theta}^{-1}(i)} \cdot \gamma$, the payment function is $P_i(\mathbf{b}) = \alpha_{\pi_{\mathbf{b},\theta}^{-1}(i)} \cdot \gamma \cdot \text{cpc}_{i\pi_{\mathbf{b},\theta}^{-1}(i)}(\mathbf{b}; \theta)$ and the utility function is: $U_i(\mathbf{b}; v_i) = \alpha_{\pi_{\mathbf{b},\theta}^{-1}(i)} \cdot \gamma_i \cdot \left(v_i - \text{cpc}_{i\pi_{\mathbf{b},\theta}^{-1}(i)}(\mathbf{b}; \theta)\right)$.

**Data Analysis** We applied our analysis to the BingAds sponsored search auction system. We analyzed eleven phrases from multiple thematic categories. For each phrase we retrieved data of auctions for the phrase for the period of a week. For each phrase and bidder that participated in the auctions for the phrase we computed the allocation curve by simulating the auctions for the week under any alternative bid an advertiser could submit (bids are multiples of cents).

See Figure 4 for the price-per-unit allocation curves $\tilde{x}_i(\cdot) = \tau_i^{-1}(\cdot)$ for a subset of the advertisers for a specific search phrase. We estimated the *average threshold* $\widehat{T}_i$ for each bidder by numerically

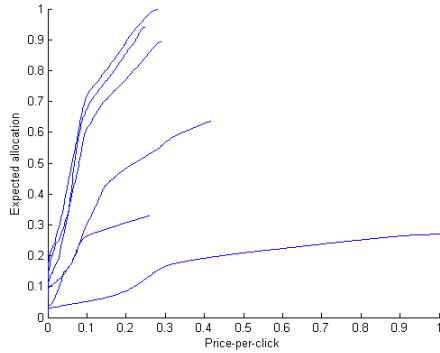

| | $\hat{\mu} = \frac{\hat{\mathbf{T}}}{\widehat{\text{REV}}}$ | $\frac{1}{\text{DPoA}} = \frac{1-e^{-\hat{\mu}}}{\hat{\mu}}$ |
|---|---|---|
| phrase1 | .511 | .783 |
| phrase2 | .509 | .784 |
| phrase3 | 2.966 | .320 |
| phrase4 | 1.556 | .507 |
| phrase5 | .386 | .829 |
| phrase6 | .488 | .791 |
| phrase7 | .459 | .802 |
| phrase8 | .419 | .817 |
| phrase9 | .441 | .809 |
| phrase10 | .377 | .833 |
| phrase11 | .502 | .786 |

**Figure 4:** (left) Examples of price-per-unit allocation curves for a subset of six advertisers for a specific keyword during the period of a week. All axes are normalized to 1 for privacy reasons. (right) Distributional Price of Anarchy analysis for a set of eleven search phrases on the BingAds system.

integrating these allocation curves along the $y$ axis. We then applied the approach described in Section 3 for each of the search phrases, computing the quantity $\hat{\mathbf{T}} = \max_{x \in \mathcal{X}} \sum_{i \in [n]} \hat{T}_i \cdot x_i = \max_{m(\cdot)} \sum_i \hat{T}_i \cdot \gamma_i \cdot \alpha_{m^{-1}(i)}$. The latter optimization is simply the optimal assignment problem where each player's value-per-click is $\hat{T}_i$ and can be performed by greedily assigning players to slots in decreasing order of $\hat{T}_i$. We then estimate the expected revenue by the empirical revenue $\widehat{\text{REV}}$.

We portray our results on the estimate $\hat{\mu} = \frac{\hat{\mathbf{T}}}{\widehat{\text{REV}}}$ and the implied bound on the distributional price of anarchy for each of the eleven search phrases in Table 4. Phrases are grouped based on thematic category. Even though the worst-case price of anarchy of this auction is unbounded (since scores $s_i$ are not equal to qualities $\gamma_i$, which is required in worst-case POA proofs [1]), we observe that empirically the price of anarchy is very good and on average the guarantee is approximately $80\%$ of the optimal. Even if $s_i = \gamma_i$ the worst-case bound on the POA implies guarantees of approx. $34\%$ [1], while the DPoA we estimated implies significantly higher percentages, portraying the value of the empirical approach we propose.

## Footnotes

[1]Even for two bidders with uniformly distributed values $U[0, a]$ and $U[0, b]$, the equilibrium strategy requires solving a complex system of partial differential equations, which took several years of research in economics to solve (see [15, 7])

[2]The theory can be easily extended to allow for different maximum achievable allocations by each player, by simply integrating the average threshold only up until the largest such allocation.

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
