[Supplementary Material]

# Supplementary material for
# " Welfare Guarantees from Data "

## A  Supplementary Figures

## B  Omitted Proofs from Section 3

**Theorem 2 (restatement)** *For any auction $A$ in a single dimensional setting and for any bid distribution $\mathcal{D}$, the distributional price of anarchy is bounded by* $\text{DPoA}(\mathcal{D}) \leq \frac{\mu(\mathcal{D})}{1-e^{-\mu(\mathcal{D})}}$, *where* $\mu(\mathcal{D}) = \frac{\max_{x \in \mathcal{X}} \sum_{i=1}^{n} T_i \cdot x_i}{\text{REV}(\mathcal{D})}$.

*Proof.* Our proof is a based on a data-dependent analog of the value and revenue covering framework of [4]. First we show that even without having distributional knowledge, the threshold functions are related to the equilibrium utility of a bidder and any target utility at any Bayes-Nash equilibrium. Specifically, either the utility of a bidder at a Bayes-Nash is high compared to his value or the average threshold $T_i$ is high.

**Lemma 8** (Value Covering). *For any bidder $i$ with value $v_i$, for any allocation amount $x \in [0, 1]$ and for any $\mu \geq 1$,*

$$u_i(v_i) + \frac{1}{\mu}T_i \cdot x_i \geq \frac{1 - e^{-\mu}}{\mu}v_i \cdot x_i. \tag{19}$$

*where $u_i(v_i) = u_i(\sigma_i(v_i); v_i)$.*

*Proof.* The proof proceeds analogously to the proof of value covering in [4]. For simplicity of notation we drop the subscript $i$, as we are focusing on a single agent and some threshold function $\tau(\cdot)$. Observe that since a player is at equilibrium it must be that for any target expected allocation $z$ he does not want to deviate to a bid that corresponds to a price-per-unit $\text{ppu}(b) = \tau(z)$, which would yield him expected allocation at least $z$:

$$u(v) \geq z \cdot (v - \tau(z)) \implies \tau(z) \geq v - \frac{u(v)}{z} \tag{20}$$

Moreover, in any case $\tau(z) \geq 0$, by definition. Thus if we define $\underline{\tau}(z) = \max(0, v - u(v)/z)$, then we have $\tau(z) \geq \underline{\tau}(z)$ and hence $T \geq \underline{T} = \int_0^1 \underline{\tau}(z)dz$.

Evaluating the integral gives $\underline{T} = v - u(v) + u(v)\log\frac{u(v)}{v}$. Thus

$$u(v) + \frac{1}{\mu}\underline{T} = u(v) + \frac{1}{\mu}\left(v - u(v) + u(v)\log\frac{u(v)}{v}\right)$$

and by dividing over by $v$:

$$\frac{u(v) + \frac{1}{\mu}\underline{T}}{v} = \frac{u(v)}{v} + \frac{1}{\mu}\left(1 - \frac{u(v)}{v} + \frac{u(v)}{v}\log\frac{u(v)}{v}\right) \tag{21}$$

The right side of Equation (21) is convex in $\frac{u(v)}{v}$, so we can minimize it by taking first-order conditions of the quantity $y + \frac{1}{\mu}\left(1 - y + y\log y\right)$ with respect to variable $y$, giving

$$0 = 1 + \frac{1}{\mu}\log y \implies y = e^{-\mu}.$$

Leading to a minimum value of that quantity of $\frac{1-e^{-\mu}}{\mu}$. Thus the right side of Equation (21) is at least this quantity, giving our desired result,

$$\frac{u(v) + \frac{1}{\mu}\underline{T}}{v} \geq \frac{1 - e^{-\mu}}{\mu}.$$

The Lemma follows by the fact that $T \geq \underline{T}$ and $x \in [0, 1]$, which allows us to multiply and divide the fraction by $x$ and then remove the $x$ in front of the quantity $u(v)$. $\square$

Given the value covering lemma we now proceed to proving the Theorem. Let $\mathbf{X}^*(\mathbf{v})$ be the welfare optimal allocation rule for valuation profile $\mathbf{v}$, i.e. the one that solves the optimization problem $\max_{x \in \mathcal{X}} \sum_{i=1}^{n} v_i \cdot x_i$. Applying the value covering inequality of Equation (19) with respect to the optimal allocation quantity $X_i^*(\mathbf{v})$ gives that for each bidder $i$ with value $v_i$,

$$u_i(v_i) + \frac{1}{\mu} T_i \cdot X_i^*(\mathbf{v}) \geq \frac{1 - e^{-\mu}}{\mu} v_i \cdot X_i^*(\mathbf{v}). \tag{22}$$

The quantity $v_i \cdot X_i^*(\mathbf{v})$ is exactly agent $i$'s expected contribution to the welfare of the optimal auction. Moreover, by the definition of $\mu(\mathcal{D})$:

$$\mu(\mathcal{D}) \cdot \text{Rev} \geq \max_{x \in \mathcal{X}} \sum_{i=1}^{n} T_i \cdot x_i \geq \mathbf{E}_{\mathbf{v}} \left[ \sum_i T_i \cdot X_i^*(\mathbf{v}) \right] \tag{23}$$

Let $\text{Util}$ denote the expected equilibrium total utility of the bidders in the auction. By Equations (22) and (23) we obtain:

$$\begin{aligned}
\text{Util} + \text{Rev} &\geq \mathbf{E}_{\mathbf{v}} \left[ \sum_i u_i(v_i) \right] + \mathbf{E}_{\mathbf{v}} \left[ \sum_i \frac{1}{\mu(\mathcal{D})} T_i \cdot X_i^*(\mathbf{v}) \right] \\
&= \sum_i \mathbf{E}_{\mathbf{v}} \left[ u_i(v_i) + \frac{1}{\mu(\mathcal{D})} T_i \cdot X_i^*(\mathbf{v}) \right] \\
&\geq \sum_i \mathbf{E}_{\mathbf{v}} \left[ \frac{1 - e^{-\mu(\mathcal{D})}}{\mu(\mathcal{D})} v_i \cdot X_i^*(\mathbf{v}) \right] = \frac{1 - e^{-\mu(\mathcal{D})}}{\mu(\mathcal{D})} \text{Opt}(\mathbf{F})
\end{aligned}$$

Since $\text{Welfare}(\sigma; \mathbf{F}) = \text{Util} + \text{Rev}$, we have our desired result:

$$\text{Welfare}(\sigma; \mathbf{F}) \geq \frac{1 - e^{-\mu(\mathcal{D})}}{\mu(\mathcal{D})} \text{Opt}(\mathbf{F}).$$

$\square$

## C  Omitted Proofs from Section 5

We begin by showing convergence of $\hat{T}_i$ to $T_i$ and $\hat{\mathbf{T}}$ to $\mathbf{T}$.

**Lemma 9** (Bounding Estimated Average Thresholds). *Suppose that the premises of Lemma 5 hold and that the function $p_i(x_i^{-1}(\cdot))$ is $L$-Lipschitz continuous. Then for each player $i$ with probability $1 - \delta$:*

$$|\hat{T}_i - T_i| \leq \tilde{O} \left( \max\{L, H\} \sqrt{\frac{H \log(1/\delta)}{T}} \right) \tag{24}$$

*Proof.* Since we focus on a single player $i$, we drop index $i$ and denote $\tau(\cdot), p(\cdot), x(\cdot)$ for $\tau_i(\cdot), p_i(\cdot), x_i(\cdot)$ and similarly for their estimated quantities. Recall that $\tau(z) = \inf_{x(b) \geq z} \frac{p(b)}{x(b)}$ and $\hat{\tau}(z) = \inf_{\hat{x}(b) \geq z} \frac{\hat{p}(b)}{\hat{x}(b)}$. Moreover, we denote with $\epsilon_x = \sup_{b \in \mathcal{B}} |\hat{x}(b) - x(b)|$ and $\epsilon_p = \sup_{b \in \mathcal{B}} |\hat{p}(b) - p(b)|$, the uniform errors on the payment and allocation curve, which be the assumptions of the theorem are upper bounded, with probability $1 - \delta$, by $\tilde{O} \left( \sqrt{\frac{H \log(1/\delta)}{T}} \right)$.

Our goal is to bound the quantity:

$$|\hat{T}_i - T_i| = \left| \int_0^1 (\hat{\tau}(z) - \tau(z)) dz \right| \leq \int_0^1 |\hat{\tau}(z) - \tau(z)| dz$$

By individual rationality we have that $p(b) \leq H x(b)$. Thus we get that $0 \leq \tau(z), \hat{\tau}(z) \leq H$ and therefore $|\hat{\tau}(z) - \tau(z)| \leq H$. Hence:

$$|\hat{T}_i - T_i| \leq \int_0^{2\epsilon_x} |\hat{\tau}(z) - \tau(z)| \, dz + \int_{2\epsilon_x}^1 |\hat{\tau}(z) - \tau(z)| \, dz \leq 2H\epsilon_x + \underbrace{\int_{2\epsilon_x}^1 |\hat{\tau}(z) - \tau(z)| \, dz}_{A}$$

It remains to bound quantity $A$. We consider any $z \in [2\epsilon_x, 1]$. By the definition of $\tau$ and $\hat{\tau}$, we obtain

$$|\hat{\tau}(z) - \tau(z)| = \left| \inf_{\hat{x}(b) \geq z} \frac{\hat{p}(b)}{\hat{x}(b)} - \inf_{x(b) \geq z} \frac{p(b)}{x(b)} \right|$$

$$\leq \underbrace{\left| \inf_{\hat{x}(b) \geq z} \frac{\hat{p}(b)}{\hat{x}(b)} - \inf_{\hat{x}(b) \geq z} \frac{p(b)}{x(b)} \right|}_{C} + \underbrace{\left| \inf_{\hat{x}(b) \geq z} \frac{p(b)}{x(b)} - \inf_{x(b) \geq z} \frac{p(b)}{x(b)} \right|}_{D}.$$

We now upper bound separately the two terms $C$ and $D$.

*Bounding $C$.* For term $C$ we have:

$$C \leq \sup_{\hat{x}(b) \geq z} \left| \frac{\hat{p}(b)}{\hat{x}(b)} - \frac{p(b)}{x(b)} \right| = \sup_{\hat{x}(b) \geq z} \frac{1}{\hat{x}(b)x(b)} \cdot |\hat{p}(b)x(b) - p(b) \cdot \hat{x}(b)|$$

$$= \sup_{\hat{x}(b) \geq z} \frac{1}{\hat{x}(b)x(b)} \cdot |\hat{p}(b)x(b) - p(b)x(b) + p(b)x(b) - p(b) \cdot \hat{x}(b)|$$

$$\leq \sup_{\hat{x}(b) \geq z} \frac{1}{\hat{x}(b)} \cdot |p(b) - \hat{p}(b)| + \sup_{\hat{x}(b) \geq z} \frac{p(b)}{\hat{x}(b)x(b)} \cdot |x(b) - \hat{x}(b)|$$

$$\leq \frac{1}{z}\epsilon_p + \frac{1}{z} \sup_{\hat{x}(b) \geq z} \frac{p(b)}{x(b)} \cdot |x(b) - \hat{x}(b)|$$

Since $z \geq 2\epsilon_x$, we have that for any $b$, with $\hat{x}(b) \geq z$, it must also be that: $x(b) \geq \hat{x}(b) - \epsilon_x \geq z - \epsilon_x > 0$, which implies that $\frac{p(b)}{x(b)} \leq H$ (by individual rationality). Which leads to the bound:

$$C \leq \frac{1}{z}\epsilon_p + \frac{H}{z}\epsilon_x \tag{25}$$

*Bounding $D$.* For quantity $D$, we proceed as follows. Let $Z = \{x(b) : x(b) \geq z\}$ and $\hat{Z} = \{x(b) : \hat{x}(b) \geq z\}$ (note that in the second set, we still use $x(b)$ to define the possible allocations, and only the set of bids is defined based on the estimated allocation function). Then:

$$D = \left| \inf_{\hat{x}(b) \geq z} \frac{p(b)}{x(b)} - \inf_{x(b) \geq z} \frac{p(b)}{x(b)} \right| = \left| \inf_{t \in \hat{Z}} \frac{p(x^{-1}(t))}{t} - \inf_{t \in Z} \frac{p(x^{-1}(t))}{t} \right|,$$

By the fact that the function $p(x^{-1}(t))$ is $L$-Lipschitz, we can bound the derivative of the function $Q(t) = \frac{p(x^{-1}(t))}{t}$ by:

$$|Q'(t)| = \left| \frac{(p(x^{-1}(t))'}{t} - \frac{p(x^{-1}(t))}{t^2} \right| \leq 2\max\left\{ \frac{L}{t}, \frac{p(x^{-1}(t))}{t^2} \right\}$$

Observe that $p(x^{-1}(t))$ is the expected payment required to get an expected allocation of $t$. By individual rationality, for any $t > 0$, the latter is at most $H \cdot t$. Thus:

$$|Q'(t)| \leq \frac{2\max\{L, H\}}{t} \tag{26}$$

Observe that for any $t \in Z \cup \hat{Z}$, $t \geq z - \epsilon_x \geq \frac{z}{2} > 0$. Hence, the function $Q(t)$ is $\frac{4\max\{L,H\}}{z}$-Lipschitz in $Z \cup \hat{Z}$. Moreover, observe that for any $\hat{t} \in \hat{Z}$, there exists $t \in Z$: $|t - \hat{t}| \leq \epsilon_x$. Hence, the two infima in expression $D$ can defer by at most:

$$D \leq \frac{4\max\{L, H\}}{z}\epsilon_x \tag{27}$$

*Concluding.* Thus we can bound quantity $A$ by:

$$A \leq \int_{2\epsilon_x}^{1} \frac{\epsilon_p + 5\max\{L, H\}\epsilon_x}{z} dz \leq \log\left( \frac{1}{2\epsilon_x} \right) (\epsilon_p + 5\max\{L, H\}\epsilon_x)$$

Combining all the above, we conclude that:

$$|\hat{T}_i - T_i| \leq 2H\epsilon_x + \log\left(\frac{1}{2\epsilon_x}\right)(\epsilon_p + 5\max\{L,H\}\epsilon_x) \tag{28}$$

Since with probability $1 - \delta$, both $\epsilon_x$ and $\epsilon_p$ are of $\tilde{O}\left(\sqrt{\frac{H\log(1/\delta)}{T}}\right)$, we get that with the same probability:

$$|\hat{T}_i - T_i| \leq \tilde{O}\left(\max\{L,H\}\sqrt{\frac{H\log(1/\delta)}{T}}\right) \tag{29}$$

since the quantity $\log(1/\epsilon_x)$, can only introduce $\log(T)$ factors in the RHS of Equation (28). $\qquad\square$

**Lemma 10** (Bounding Estimated Optimal Threshold Quantity). *Let* $\hat{\mathbf{T}} = \max_{x \in \mathcal{X}} \sum_{i=1}^{n} \hat{T}_i \cdot x_i$. *Then with probability* $1 - \delta$:

$$|\hat{\mathbf{T}} - \mathbf{T}| \leq \tilde{O}\left(n\max\{L,H\}\sqrt{\frac{H\log(n/\delta)}{T}}\right) \tag{30}$$

*Proof.* By Lemma 9 and a union bound across players, we have that with probability $1 - \delta$:

$$\sup_{i \in [n]} |\hat{T}_i - T_i| \leq \tilde{O}\left(\max\{L,H\}\sqrt{\frac{H\log(n/\delta)}{T}}\right) \tag{31}$$

Moreover, for any allocation $x \in \mathcal{X}$:

$$\left|\sum_i (\hat{T}_i - T_i)x_i\right| \leq \sum_i |\hat{T}_i - T_i| \leq \tilde{O}\left(n\max\{L,H\}\sqrt{\frac{H\log(n/\delta)}{T}}\right) \tag{32}$$

Thus we can bound the error in $\mathbf{T}$ as:

$$\left|\hat{\mathbf{T}} - \mathbf{T}\right| = \left|\max_{x \in \mathcal{X}} \sum_i (\hat{T}_i \cdot x_i - \max_{x \in \mathcal{X}} \sum_{i \in [n]} T_i x_i\right| \leq \sup_{x \in \mathcal{X}} \left|\sum_i (\hat{T}_i - T_i)x_i\right|$$

$$\leq \tilde{O}\left(n\max\{L,H\}\sqrt{\frac{H\log(n/\delta)}{T}}\right)$$

The latter concludes the proof of the theorem. $\qquad\square$

We are now ready to show our main estimation theorem.

**Theorem 7 (Restatement).** *Under Assumption 6 and the premises of Lemma 5, with probability* $1 - \delta$:

$$\frac{\text{OPT}(\mathbf{F})}{\text{WELFARE}(\sigma;\mathbf{F})} \leq \frac{\hat{\mu}}{1 - e^{-\hat{\mu}}} + \tilde{O}\left(n\max\{L,H\}\sqrt{\frac{H\log(n/\delta)}{T}}\right) \tag{33}$$

*Proof.* First note that $\widehat{\text{REV}} = \frac{1}{T}\sum_{t=1}^{T} \mathbf{b}^{i:T}$. Thus, using standard Hoeffding's inequality we obtain that

$$P\left(\left|\widehat{\text{REV}} - \text{REV}\right| > \tau\right) \leq 2e^{-\frac{2T\tau}{(nH)^2}}.$$

Thus with probability at least $1 - \delta/2$

$$\left|\widehat{\text{REV}} - \text{REV}\right| \leq nH\sqrt{\frac{\log(4/\delta)}{2T}}.$$

Combining this result with the result of Lemma 10, we find that with probability at least $1 - \delta$

$$|\hat{\mu} - \mu| \leq \tilde{O}\left(n\max\{L,H\}\sqrt{\frac{H\log(n/\delta)}{T}}\right).$$

Let $\widehat{\rho} = \widehat{\mu} \left( 1 - e^{-\widehat{\mu}} \right)^{-1}$. Since the function $f(x) = x/(1 - e^{-x})$ is Lipschitz we can conclude that

$$|\widehat{\rho} - \rho| \leq \tilde{O} \left( n \max\{L, H\} \sqrt{\frac{H \log(n/\delta)}{T}} \right)$$

with probability at least $1 - \delta$. $\qquad\square$