[Reviews · NeurIPS 2017]

Reviewer 1



The paper introduces a way to provide data dependent bounds on the price of anarchy (PoA). The main idea behind the paper is to provide a finer notion of the PoA restricted to a particular bid distribution. The authors show that we can upper bound this quantity by a ratio of two simple expectations which can in turn be estimated from samples under some reasonable assumptions. The ideas behind the paper are simple, well explained and are really well structured. There are only a few things that would improve the paper: 1) When the authors compare their approach with that of inverting the equilibrium function, they neglect to mention that the inversion algorithm would estimate the true PoA while the proposed algorithm here only estimates an upper bound. Nevertheless, I agree with the authors that those estimates only hold under restrictive assumptions on the density function of the values. 2) Is assumption 6 absolutely necessary ? I understand that it is needed in your proof technique. However it seems that since the denominator defining \mu(D) is always larger than the numerator, a close to 0 denominator should be unlikely unless the numerator is also close to 0. 3) Line 140 I think in the upper bound it should be T_i / \mu Line 141 This should also be (1 - e^-\mu)/ \mu Ditto for eq (9) and (10)

Reviewer 2



This paper is about game theory which falls out of my expertise. I can only give the following points confidently. 1. The title does not contain much information since it ambitiously relates itself to too many branches in statistics. 2. The equation number and theorem number in the appendix should be carefully edited instead of left as question marks.

Reviewer 3



This paper studies a non-worst-case notion of the price of anarchy (PoA). An auction A’s PoA is the worst-case ratio between the welfare of the optimal auction and the welfare in a Bayes-Nash equilibrium of A, taken over all value distributions and equilibria. The authors propose a non-worst case relaxation which they call the distributional PoA (DPoA), which is defined by a distribution D over bids. It measures the same ratio as the original PoA, but only over all value distributions and equilibria that could generate the bid distribution D. They bound the DPoA for single-dimensional settings and show that it can be estimated using samples from D. In one example, the authors provide an upper bound on the DPoA for the single item first price auction (Lemma 1). There are a few things I couldn’t reproduce in the proof. In line 140, the authors write that 1/\mu*\int_0^{v_i(1 – e^{-\mu})} (1 – G_{-i}(z)) dz <= T_i. Isn’t it less than or equal to T_i/\mu? Also, it seems as though if I add 1/\mu*\int_0^{v_i(1 – e^{-\mu})} (1 – G_{-i}(z)) dz to the right-hand-side of Equation (8), I’ll get 1/\mu*\int_0^{v_i(1 – e^{-\mu})} (1 – G_{-i}(z) + G_{-i}(z)) dz = 1/\mu*\int_0^{v_i(1 – e^{-\mu})} dz = v_i(1 – e^{-\mu})/\mu. (The authors write that it’s just v_i(1 – e^{-\mu}).) Lastly, in line 145, the authors write that \sum_i T_i*x_i >= max_i T_i. Since this is a single-item auction, there’s exactly one x_i that equals 1 and the rest equal 0. Should the inequality be flipped? (\sum_i T_i*x_i <= max_i T_i.) I would really like to see the proof of Lemma 5 written out. It’s not clear to me how it follows immediately from Lemma 9.9 in [5]. Rademacher complexity does have nice compositional properties. However, I’m not convinced, for example, that “multiplication of bid vectors with constants” is covered by Lemma 9.9 in [5]. Lemma 9.9 in [5] applies to function classes mapping some domain X to R. The function class defined by “multiplication of bid vectors with constants” maps from R^{n-1} to R^{n-1}. How exactly does Lemma 9.9 apply? Also, it seems like multiplication would have to be involved in the form of many payment functions. For example, in the first price auction, the payment function is b_i*x_i, where x_i can be written as {b_i > b_1}*…*{b_i > b_{i-1}}*{b_i > b_{i+1}}*…*{b_i > b_n}. Is there a way to write this without multiplication? The authors write that the conditions in Lemma 5 hold for sponsored search auctions. Is there no multiplication in that setting either? I think this paper is currently better written for an economics community as opposed to a learning community. For example, it’s not clear to me what a pay-your-bid auction is. On line 187, the authors write that “For any auction, we can re-write the expected utility of a bid b: u_i(b, v_i) = x_i(b)(v_i – p_i(b)/x_i(b)).” What if x_i(b) = 0? I think that overall, this is a really interesting contribution to beyond-worst-case analysis and the experiments seem to validate its practical relevance. However, I think the presentation could be improved to make it more understandable for an ICML audience. I’m also hesitant to recommend it for acceptance without a proof of Lemma 5. ======after rebuttal====== I'm willing to believe Lemma 5 if the authors are assuming that number of bidders is constant. I didn't realize that they were making this assumption. The authors should mention this explicitly in the preliminaries and/or the lemma statement.